# Research on Cutting Edge form Factor of Milling Tool after Drag Finishing Preparation Based on Discrete Element Method

**Lihong Zhou [1], Yongguo Wang [1,*] and Dejin Lv [2]**

1   School of Mechatronics Engineering and Automation, Shanghai University, Shanghai 200444, China; zhou_zihai@shu.edu.cn
2   Guohong Tool System (Wuxi) Co., Ltd., Wuxi 214000, China; lvdejin2015@163.com
*   Correspondence: ygwang@shu.edu.cn

**Abstract:** Cutting edge preparation is a precision machining process that improves the surface quality of cutting tools through the relative movement of abrasives and the tool. Research on removing materials in drag finishing can be greatly beneficial to tool manufacturing. This paper proposes the hypothesis that both abrasive wear and erosion wear act on the surface of milling tools and discusses the material removal models for abrasive wear and erosion wear. The influence of immersion depth, abrasive velocity, abrasive radius, and abrasive density on the material removal rate in two material removal forms is compared and validated by discrete element simulations. The results show that immersion depth has a greater impact on abrasive wear, while abrasive properties have a greater impact on erosion wear. The correlation between simulation results and theoretical models demonstrates the sensitivity of the two forms of wear on this surface to parameter change differences. Dragging finishing was conducted to verify the effectiveness of the simulation, and the effects of immersion depth, dragging velocity, and abrasive properties on the edge radius and form factor (K value) were studied.

**Keywords:** cutting edge preparation; material removal; discrete element simulations; erosion wear; abrasive wear



## 1. Introduction

The process of cutting edge preparation by dragging finishing refers to a finishing process in which the clamped tools are immersed in an abrasive pool and kept moving, which is a necessary finishing process in the manufacturing of milling tools and one of the methods in abrasive flow finishing [1]. After abrasive flow impacts the cutting edge, it eliminates surface micro-cracks and removes irregular burrs formed during the grinding process, resulting in a uniformly arc-shaped edge cross-section. Extensive research findings indicate that the cutting forces and surface integrity are considerably affected by the tool's edge radius [2,3]. Previous research on tool preparation can be divided into two main areas: the comparison of different methods and the influence of different process parameters.

Common methods for preparing cutting edges include basic abrasive jets, brushing, drag finishing (DF), and grinding wheel methods. Wang et al. compared the effects of three preparation methods, namely abrasive jets, brushing, and DF, on hard alloy blade performance [4]. Denkena conducted a comparison between the effects of brushing and grinding wheel techniques on the results of preparing hard alloy tools [5]. These comparative results hold significant guidance for the preparation processes of specific tools. Furthermore, the study of process optimization is extensively involved. Wang et al. employed the air wet abrasive jet machining (PAWAJM) technique, which yielded tool blades with elevated lifespan, hardness, and residual compressive stress in the vicinity of the tool edge when compared to untreated instruments [6]. Aurich et al. used elastic bonded super-abrasive grinding wheels, accurately producing symmetrical edge profiles with significantly greater efficiency than traditional finishing methods [7,8]. Guan et al.

put forward a method for the preparation of a tool edge with magnetorheological fluid that contains abrasives. Through the control of the magnetic field's intensity, this process is capable of generating diverse radii in distinct edge locations [9]. Each of these novel preparation methods has its individual advantages in the production process of milling tools, which provide a greater array of technical options in the actual machining process.

DF is the most widely adopted edge preparation process due to its lower cost. A significant amount of relevant research has also been conducted in this area. Lv et al. undertook a comparative study of the DF outcomes on cemented carbide milling tools using different materials [10]. F. Pérez-Salinas et al. conducted experimental research on broach DF and utilized artificial neural networks to predict the tool edge radius [11]. Hashimoto et al. measured the contact force on the workpiece during gyroscopic finishing and analyzed the impact of several factors, including the workpiece's immersion depth, on the contact force [12].

The above studies mainly focus on the impact of various processes and parameters on the radius of the cutting edge and form-factor (K value). However, these direct studies into process parameters have limitations. The applicability of research results can only be extended if there is a complete understanding of the wear mechanism of milling tool edge preparation. As a component of abrasive flow finishing, the theoretical examinations of DF could refer to other abrasive flow finishing methods. The abrasive flow finishing is the application of numerous abrasives to the workpiece material to achieve material removal via relative collisions, with theoretical studies focusing on abrasive and erosion wear along with establishing mathematical models for material removal.

Abrasive wear refers to the process by which particles are pressed against the surface of a material and generate a ploughing action [13]. Ohlert et al. conducted an analysis of the material removal mechanism in gyroscopic finishing and developed an empirical model to evaluate the quality of material removal based on contact force, contact frequency, relative velocity, and process conditions [14]. Barletta et al. introduced a novel technique for achieving superior surface quality in workpieces through fluidized bed-assisted DF. The authors integrated the concept of localized plastic deformation and energy absorption methods in their investigation of material removal in plastic materials with fluidized bed finishing [15,16]. Azami et al. proposed a novel theoretical model for predicting surface roughness after gyroscopic finishing, factoring in parameters like abrasive particle size, velocity, and gap distance [17].

Erosive wear refers to the process of abrasive particles impacting the surface of the material at a certain angle and moving part of the material away [18,19]. Gao and colleagues used spherical magnetic abrasives for magnetic flow finishing, applying abrasive wear theory and conducting experimental analysis to reduce surface roughness and enhance the material removal rate [20]. Yang and colleagues investigated the impact of $SiO_2$ and $Al_2O_3$ particles on the erosion wear of titanium alloy. They developed the Tabakoff erosion wear model, established model parameters, and analyzed the wear patterns and erosion rates of both particles on titanium alloy [21]. Yaer et al. employed finite element simulation to investigate the particle erosion wear process of high-temperature alloy materials. They analyzed wear removal at varying angles and speeds and observed microstructural and hardness changes on the material surface subsequent to erosion [22].

The aforementioned literary survey has comprehensively investigated material removal rate models in abrasive flow finishing and has demonstrated their efficacy through experimentation. The above studies reveal that material removal in abrasive flow finishing is the result of combined abrasive and erosion wear. Nonetheless, all the above studies only applied to wear on simple regular surfaces. There exists a research gap with regard to the material removal mechanism of components with special shapes such as milling tools. During the process of DF, the milling tool experiences distinctly different movements of abrasives on the rake and flank face near the edge due to its unique shape. Consequently, it is imperative to analyze each surface of the tool separately.

Due to the complex geometry of the milling tool, drag finishing results in different distributions of material removal rates on the rank and flank faces, which are caused by their differing macro-collision characteristics. This article aims to study the material removal mechanism and its macroscopic behavior in dragging finishing. The material removal models for abrasive wear and erosion wear that take place during DF are discussed. Based on the simulation of the DF process employing EDEM, the motion status of abrasives near the milling tool and the cumulative energy of different wear forms on the milling tool's surface are observed. This study compares and analyzes the rates and types of material removal at various positions near the edge, with a combination of theoretical and simulation findings. Verification experiments are designed according to an orthogonal design, with the resultant data used to investigate the impacts of dragging velocity, immersion depth, and abrasive properties on the edge radius and K value.

## 2. Removal Mechanism Analysis of Abrasive Finishing Materials

The motion path of the milling tool is a secondary planetary motion in the DF process. The material removal mechanism involves a combination of erosion wear and abrasive wear caused by the relative collision between the abrasive and the milling tool, which are shown in Figure 1. The mechanisms of these two wear forms are combined to finally form the milling cutting edge.

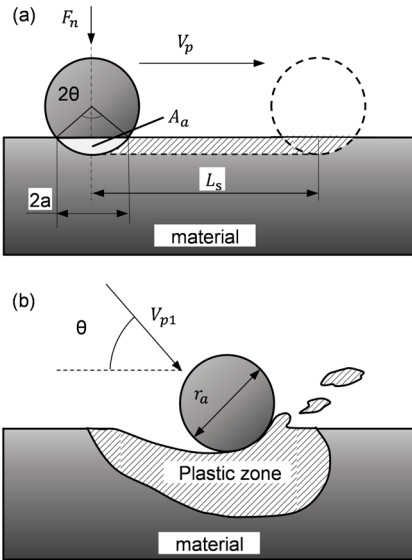

**Figure 1.** Wear models: (**a**) abrasive wear model and (**b**) impact erosion wear model.

### 2.1. Material Removal Model of Abrasive Wear

The process of abrasive wear occurs when abrasive grains move at low velocities. As shown in Figure 1a, an abrasive is pressed into the milling tools material by the pressure of surrounding abrasive grains and accompanied by a ploughing effect. Due to the irregular shape of the abrasive surface, including ridges and sharp corners, the material can be removed even if the radius of the abrasive is larger than the depth of the indentation. As the occurrence of abrasive wear is the result of a large number of abrasive grains acting simultaneously, simplification of this process is necessary for relative study. Thus, the following assumptions are made:

1. The size and material of abrasive grains as well as the applied load are all identical.
2. The surface material of the tool is uniform and can be treated as a smooth curve.
3. Ignore the deformation of abrasive particles during wear.

It can be deduced that the area of the indentation produced by the action of a single particle on the plane is as shown in Figure 1a and the cross-sectional area $A_a$ of groove generated can be easily calculated based on geometry.

$$A_a = \frac{1}{2}r_a^2(2\theta - sin2\theta) \tag{1}$$

where $r_a$ is the radius of abrasive and $\theta$ is the indentation angle. Assuming that the number of abrasive particles undergoing effective wear at the same time in the unit area is $N_s$, the mass of material removal $M_0$ is:

$$M_0 = \frac{1}{2}\rho N_s V_a T r_a^2(2\theta - sin2\theta) \tag{2}$$

where $\rho$ is the material density, $T$ is the dragging time, and $V_a$ is the relative velocity between the abrasive grain and the milling tool and it is the number of abrasives acting simultaneously on the surface per unit area. This model has been widely accepted since the last century, and a large number of research results have continuously improved upon it [23]. Model improvements are as follows.

Considering the coexistence of $\theta$ and $sin\theta$ in Equation (2), it is difficult to compare the influence of normal pressure and average abrasive radius on the volume of wear. Therefore, $t$ is introduced to represent the sine value of the indentation angle.

$$t = sin(\theta) = \frac{a}{r_a} = \frac{1}{r_a}\sqrt{\frac{F_n}{H_w \pi}} \tag{3}$$

where $F_n$ is normal force and $H_w$ is the hardness of the tool material,

$$M_0 = \rho N_s V_a T r_a^2 \left( \arcsin(t) - t\sqrt{1-t^2} \right) \tag{4}$$

Assuming $\bar{t}$ is the value of $t$ when the average indentation depth is the indentation angle, and substituting it into the above equation, the second-order Lagrange polynomial expansion of $sin(t)$ at $\bar{\theta}$ yields the following result,

$$M_0 \approx \rho N_s V_a T r_a^2 \left( k_1 t^2 + k_2 t + k_3 \right) \tag{5}$$

Simultaneous Equations (3) and (5),

$$M_0 \approx \rho N_s V_a T \left( k_1 \frac{F_n}{H_w \pi} + k_2 r_a \sqrt{\frac{F_n}{H_w \pi}} + k_3 r_a^2 \right) \tag{6}$$

where $k_1$, $k_2$, and $k_3$ are coefficients related to the average indentation depth, and simple mathematical deduction shows that they are all positive values. Consequently, both the abrasive grain radius and the radius of the indentation circle have a positive impact on the material removal during abrasive wear. Considering the relationship between the radius of the indentation circle and the normal pressure, the normal force also has a positive effect.

When the milling tool material is determined, the material removal rate is closely related to the relative motion speed of the abrasive grains and normal force exerted on the abrasive grains and abrasive grain radius.

Malkorra et al. supposed the abrasive grains are a continuous fluid during the abrasive flow finishing and conducted simulations. The immersion depth of the milling tool and relative equation were discussed [24].

$$p = \rho_a A_b h g \tag{7}$$

where $\rho_a$ is the abrasive density, $A_b$ is the maximum cross-sectional area of the abrasive grain, $h$ is the immersion depth at the location of the abrasive grain, and $g$ is the gravity acceleration.

$$F_n \propto \rho_a A_b h \tag{8}$$

This means that the normal pressure imposed on the abrasive particle is positively correlated with the product of the abrasive particle density, maximum cross-sectional area of the abrasive grain, and immersion depth. Considering the above, the material removal of abrasive wear is closely related to the properties of the abrasive grains (such as size and density), the relative velocity between the abrasive grains and the milling tool, and the immersion depth in DF.

### 2.2. Material Removal Model of Impact Erosion Wear

In addition to abrasive wear in which abrasive particles are pressed to produce scratches on the surface, the process of abrasive particles directly impacting the material surface from a certain angle and taking away part of the material is also a major material removal model in abrasive flow finishing, which is called erosion wear as shown in Figure 1b. For a long time, a lot of research according to the impact angle, plasticity of materials, and erosion rate has been conducted and there are various models, such as the stress fatigue fracture model, brittle fracture model, micro-cutting model, and E/CRC empirical model. Considering that each particle is subjected to the force of the surrounding particles in DF, which is different from when a wear particle has residual kinetic energy and quickly leaves the surface of the material after erosion, the erosion wear model of stress-fatigue fracture is selected, which is based on the conservation of energy in the impacting process. In the same way, the model is simplified by the following assumptions:

1. The influence of plastic deformation and cyclic residual stress can be ignored under low velocity erosion conditions.
2. Ignore the energy and heat loss of abrasive particles when they impact the material.
3. The whole process of stress fatigue erosion is irreversible deformation.

According to the above assumptions, all energy loss before and after particle impact acts on the removal of the milling tool material. The velocity change before and after particle impact can be calculated according to the empirical formula proposed by Tabakoff et al. [21].

$$\frac{V_a}{V_a'} = V_a f^2(\theta_1) \tag{9}$$

where $V_a'$ is the velocity after erosion of abrasive particles and $f(\theta_1)$ is a function of the erosion angle. Thus, the energy loss after the impact is,

$$\Delta E_k = \frac{2}{3}\pi\rho_a r_a{}^3 V_a{}^2\left(1 - f^2(\theta_1)\right) \tag{10}$$

where $\Delta E_k$ is the amount of kinetic energy changed after the impact of a particle.

Assuming that the energy required to remove a unit volume of material is $\varepsilon$, since the energy required to remove a unit volume of material is proportional to the erosion hardness $E_t$,

$$\varepsilon \propto \frac{H_w{}^2}{E_t} \tag{11}$$

Combined with Equations (10) and (11), Erosion rate $E_d$ is defined as the ratio of the quality of the material removed per unit of time and the cumulative mass of particles impacted on the surface. The relationship between $E_d$ and material hardness $H_w$ is:

$$E_d \propto \frac{E_t\rho_a r_a{}^3 V_a{}^2\left(1 - f^2(\theta_1)\right)}{H_w{}^2} \tag{12}$$

Accordingly, the definition of erosion rate and the mass of material removal $M_0$ can be calculated as follows,

$$M_0 \propto \frac{E_t\rho_a{}^2 r_a{}^6 N_e V_a{}^3\left(1 - f^2(\theta_1)\right)}{H_w{}^2} \tag{13}$$

where $N_e$ is the number of abrasive particles undergoing effective impact at the same time in unit area. It can be concluded that when the material is determined, the material removal rate of the milling tool erosion wear is related to the properties of the wear particles (density and particle size), erosion velocity, and erosion angle.

The purpose of this paper is to compare the material removal mechanisms of abrasive and erosive wear, both of which are influenced by the properties of the abrasive grain and the relative velocity between the abrasive grain and the milling tool. Abrasive wear is specifically influenced by the immersion depth. The influence of different abrasive types, immersion depth, and dragging speed on the material removal process of milling tool edge preparation during milling will be discussed. However, in the simulation, all abrasive shapes are irregular because they are difficult to quantify. Therefore, the abrasive shape is not considered and the effect of abrasive hardness, density, and size on the experimental results is investigated using different types of abrasives.

The comparison of the different parameters in the models of material removal rate for abrasive wear and erosive wear in Equations (6) and (13) can help us infer that changes in different process parameters will lead to changes in the proportion of abrasive wear and erosive wear in DF. Due to the unique shape of the milling tool and the motion pattern of the secondary planetary motion, it is reasonable to speculate that the wear forms at different positions of the milling tool are different. This point will be verified through discrete element simulation (DEM) in the next section of this paper.

## 3. DEM Simulation and Experiment of Tool Edge Preparation

### 3.1. Abrasive and Milling Tool

The simulation model in this section uses the same milling tool and abrasive properties as the experiment. The experiment utilized three types of abrasive particles with distinct characteristics: K3/400, SiC, and SiO$_2$, which were provided by the German OTEC company; the microscope images of the abrasive are shown in Figure 2. K3/400 is composed of a walnut core, which is widely used due to its cost efficiency and low wear. SiC and SiO$_2$ have higher hardness and a sharp angle for more efficient material removal, and in order to reduce the wear between the particles and make full use of the particles, some K3/400 particles are mixed in the SiC and SiO$_2$ particles. The properties of the carbide and abrasive particles are shown in Table 1.

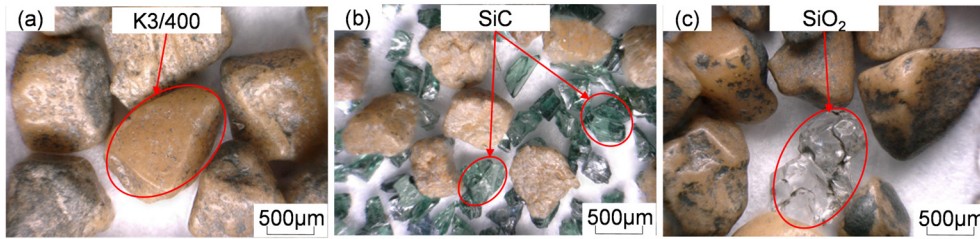

**Figure 2.** Microscope image of abrasive particles.

**Table 1.** Abrasive medium used for DF.

| Typical Analysis | Hardness [kgf/mm$^2$] | Density [g/cm$^3$] | Average Abrasive Diameter [mm] | Shear Modulus [Pa] | Poisson's Ratio | Coefficient of Restitution | Coefficient of Static Friction |
|---|---|---|---|---|---|---|---|
| Tool | 1700 | 14.5 | —— | $2.35 \times 10^{11}$ | 0.3 | 0.5 | 0.25 |
| K3/400 | 1500 | 3.0 | 0.8~1.3 | $1.2 \times 10^{11}$ | 0.1 | 0.45 | 0.2 |
| SiC | 2700 | 3.2 | 0.4~0.8 | $1.78 \times 10^{11}$ | 0.14 | 0.75 | 0.25 |
| SiO$_2$ | 2400 | 3.6 | 0.8~1.3 | $2.12 \times 10^{11}$ | 0.3 | 0.85 | 0.3 |

The experiment employed carbide milling tools produced by ANCATX7 five-axis CNC grinding machine manufactured by ANCA, Australia. The cutting edge of the tool exhibited irregular defects after grinding, which caused stress concentration and premature tool failure, which is shown in Figure 3. Microscopic damage and defects are common after grinding in milling tools, thus, the edge preparation is indispensable for all milling tools. The tool parameters are shown in Table 2. These end mills are in a four flute, measuring six diameters with hardness HV = 1.7 KN/mm$^2$, supplied by Guohong Tool System (Wuxi, China) Co., Ltd.

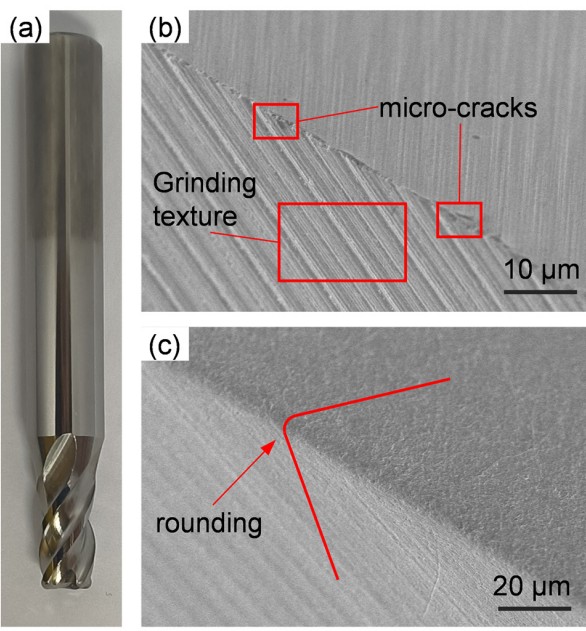

**Figure 3.** (**a**) Carbide end mills, (**b**) cutting edge micro-defect before dragging finishing, and (**c**) cutting edge micro-defect after dragging finishing.

**Table 2.** The geometric parameters of end mills.

| Parameter | Value |
|---|---|
| Shank diameter D | 8 mm |
| Tool diameter d | 6 mm |
| Number of teeth Z | 4 |
| Cutting edge length l | 10 mm |
| Helix angle β | 40° |
| Radial rake angle γ | 0° |
| Radial relief angle α | 8° |
| Corner radius R | 1 mm |

### 3.2. Discrete Element Model Setup

For observing the particle flow behavior in DF of milling tools and analyzing the wear forms at different position, this section outlines the study conducted using a simulation of the DF of the milling tool using the commercial software EDEM 2022, which cannot be directly observed in the DF experiment. The establishment of the milling tool model and discrete element particle model based on DF of the milling tool is as follows.

EDEM is a widely used discrete element software that contains various collision and thermodynamic models. The particle–particle collision model adopted in this study is the non-slip Hertz–Mindlin contact model, while the particle–material collision models used are the Archard wear model and the relative wear model, which are used to calculate the material wear volume and the cumulative wear energy. The mesh size falls within the standard range defined by EDEM, with a value set at 2.5 times the minimum particle size, resulting in a mesh number of 48,488.

The abrasive particles are simplified to the shape of a standard sphere to reduce the computational load of the simulation. The particle size distribution generated in the particle factory follows a normal distribution. All abrasive particles are placed in a hexagonal cylinder container to ensure that there is sufficient counterforce when the milling tool is dragged. To reduce the size of the model, the ineffective shank part of the milling tool is removed. The motion trajectory of the milling tool edge is simplified to a planetary motion. The milling tool revolves around the center of the material pool and rotates itself. The rotation is divided into forward and reverse based on the helix direction of the milling

tool. Figure 4a represents the velocity vector map of abrasive particles around the milling tool during the simulation and Figure 4b represents the simulation model. The condition at the position recorded in Figure 4c is taken as a result of the simulation.

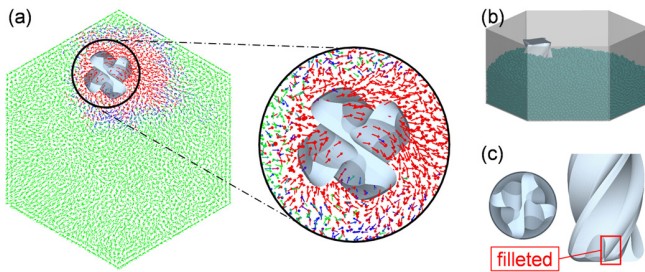

**Figure 4.** Simulation model. (**a**) Simulated velocity vector of abrasive particles. (**b**) Simulation process. (**c**) Milling tool filleted.

The simulation experiment comprises seven groups, which are all 5 s in duration. The method of controlling variables is used to investigate the effects of immersion depth, dragging velocity, and abrasive properties on the simulation results. To maintain computational efficiency, the discrete element model should not be excessively large. Therefore, parameters for milling tool edge preparation are chosen within the confines of downsizing the model. The chosen parameter settings for each are presented in Table 3.

**Table 3.** Simulation parameters and levels.

| Factors | Immersion Depth (mm) | Dragging Velocity * | Abrasives |
|---|---|---|---|
| 1 | 5 | 1.4/4 | K3/400 |
| 2 | 10 | 1.75/5 | SiC |
| 3 | 15 | 2.1/6 | $SiO_2$ |

* In dragging velocity, 1.75/5 means that the rotation speed is 5 rad/s and the revolution speed is 1.75 rad/s in dragging velocity.

### 3.3. DF Experiment Settings

This section conducts DF experiments on the milling tool after grinding. The DF machine is shown in Figure 5. Considering the edge radius limit of $SiO_2$ abrasive particles, dragging time is controlled to be 10 min in each group. The milling tool is cleaned with an ultrasound after DF. The edge radius of the milling tool corner and the K value, as well as the surface roughness of the rake face and flank faces near the edge, are observed and measured with Infinite-Focus G5 microscope by Alicona as shown in Figure 6. After three repetitions of each experimental group, the average values are taken and shown in Table 4.

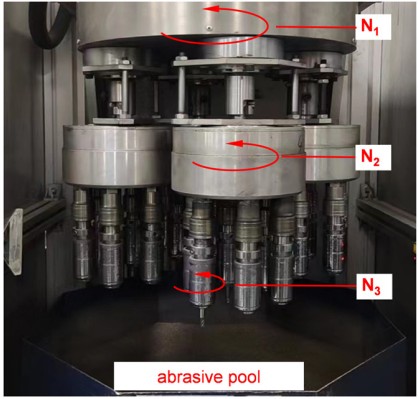

**Figure 5.** DF machine, N1, N2, and N3 represent respectively the first-order revolution of the second-order planetary motion, the second-order revolution and the rotation of the tool.

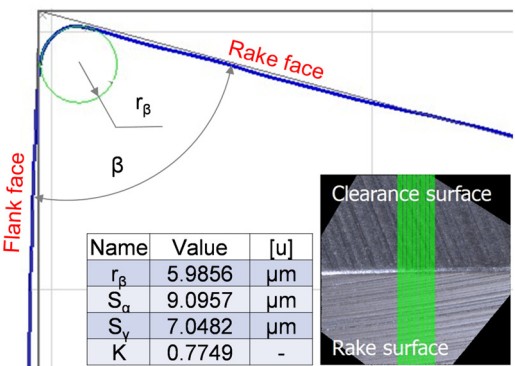

**Figure 6.** Cutting edge measurement using Alicona microscope.

**Table 4.** The radius and K value of the cutting edge in the Taguchi experiment.

| Group Number | Immersion Depth | Dragging Velocity | Abrasive Properties | Edge Radius (μm) | K Value | Rake Face Ra (μm) | Flank Face Ra (μm) |
|---|---|---|---|---|---|---|---|
| 1 | 1 | 1 | 1 | 3.097 | 0.833 | 0.254 | 0.1705 |
| 2 | 1 | 2 | 2 | 4.860 | 0.747 | 0.2485 | 0.1915 |
| 3 | 1 | 3 | 3 | 9.553 | 0.937 | 0.2645 | 0.141 |
| 4 | 2 | 1 | 2 | 5.977 | 0.860 | 0.2075 | 0.1605 |
| 5 | 2 | 2 | 3 | 11.353 | 0.940 | 0.235 | 0.126 |
| 6 | 2 | 3 | 1 | 3.880 | 0.680 | 0.2565 | 0.196 |
| 7 | 3 | 1 | 3 | 11.870 | 0.967 | 0.233 | 0.125 |
| 8 | 3 | 2 | 1 | 5.1467 | 0.737 | 0.225 | 0.154 |
| 9 | 3 | 3 | 2 | 9.970 | 0.743 | 0.2185 | 0.115 |

The representation of the K value as a significant parameter post milling tool edge preparation process indicates the specific shape of the milling tool edge cross-section, as illustrated in Figure 7. When K equals 1, the tool edge exhibits a symmetric profile, wherein $S_\gamma$ equals $S_\beta$, implying equal material removal on the tool face before and after the edge preparation process. However, when K exceeds 1, the material removal on the rake face surpasses that on the flank face, resulting in an asymmetric edge. Consequently, a reduction in the tool rake angle occurs, significantly influencing cutting forces and machining surface integrity, and vice versa. Thus, utilizing the K value to analyze the disparity in material removal between the tool faces before and after edge preparation proves to be effective.

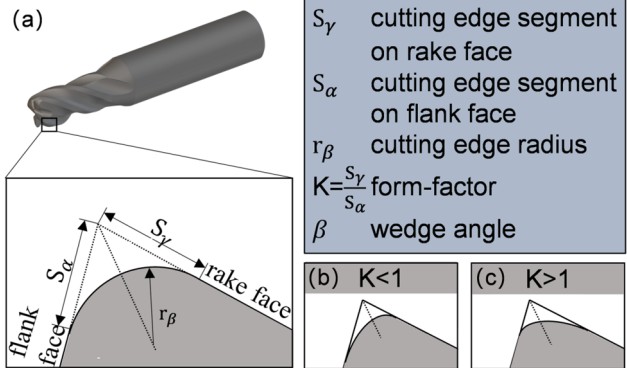

**Figure 7.** Form-factor method and the parameter used in edge geometry determination. (**a**) Characterization of tool edge parameters. (**b**) Waterfall hone (K < 1). (**c**) Trumpet hone (K > 1).

## 4. Discussion

### 4.1. Results of Simulation and Taguchi Experiment

In this section, the DEM is used to explore the wear forms of the milling tool position during the DF, combined with an analysis of the effects of different parameters on abrasive wear and erosion wear. The scalars for abrasive flow processing are the radius of the cutting edge and the K value. The validity of the simulation results is verified through experiments and the effects of each parameter on the K value are studied.

As is well known, the amount of material removal exhibited a direct correlation with the cumulative energy loss identified in the DEM [25]. Abrasive wear removal can be represented by the cumulative energy caused by the slippage of abrasive grains on the material surface, known as Tangential Cumulative Energy (TCCE). Equally, erosive wear removal can be represented by the normal cumulative energy (NCCE) [26]. However, the reduction in the model affects the size of the material pool, thereby restraining the range of variations in the immersion depth and dragging velocity of the milling tool. As a result, the comparison of TCCE and NCCE is not feasible, and subsequent analysis singularly considers the alterations in TCCE and NCCE.

To gain insight into the wear patterns near the cutting edge of the milling tool, two sets of DF simulations were conducted. These simulations included one for the forward motion and one for the reverse motion, both utilizing the median values for the parameters. Figure 8 displays the distribution of NCCE and TCCE surrounding the milling tool's rounded surface at the end of the simulation. For the reverse rotation of the milling tool, Figure 8a,b indicates that NCCE is predominant on the flank face and less present on the rake face, whereas TCCE appears mostly on the flank face and is almost absent on the rake face. Figure 8c,d depicts milling tool forward rotation, where both NCCE and TCCE occur on both the rake and flank faces. In conclusion, different wear forms near the cutting edge display varying distributions. Equations (6) and (13) show that erosion wear and abrasive wear are sensitive to different parameters, resulting in differences in subsequent macro characterization.

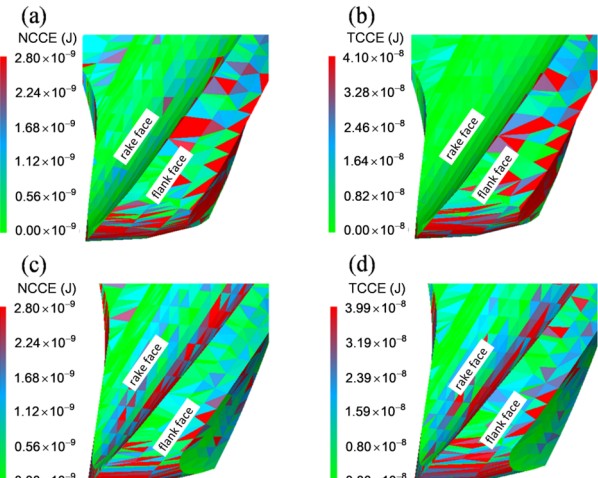

**Figure 8.** Accumulated energy of forward and reverse. (**a**) Reverse NCCE. (**b**) Reverse TCCE. (**c**) Forward NCCE. (**d**) Forward TCCE.

Figure 9 displays the findings of the surface roughness of the rake and flank face proximate to the edges of the DF experiments. The results demonstrate that the roughness of the rake edge exceeds that of the flank edge under various dragging parameters. This is consistent with the simulation outcomes, which resulted from the relative motion of the abrasive particles and the tool during the DF. These results remain independent of the process parameters. When the milling tool is in reverse, both types of wear have a greater impact on the flank face. In contrast, during forward motion, both types of wear are present

on both the rake and flank faces. As a result, the removal rate and surface finish are higher on the flank face.

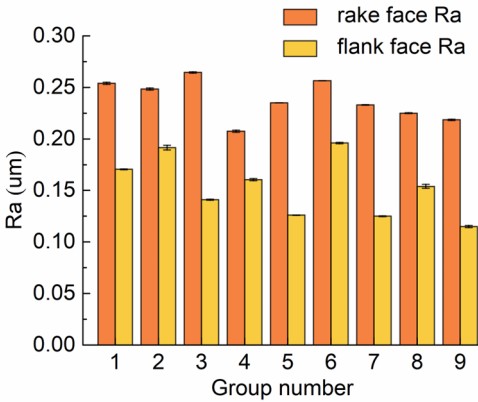

**Figure 9.** Rake face Ra and flank face Ra after DF.

By comparing the wear form distributions at various edge positions and utilizing Equations (6) and (13), it can be deduced that the material removal rate will yield different results due to dissimilar orders and coefficients of immersion depth, dragging velocity, and abrasive properties in the two equations. Figure 10 displays the cumulative normal and tangential energy produced by varying immersion depths, milling speeds, and abrasive particle types. The total energy of the milling tool is statistical based on the bisection method of the maximum total energy, ensuring that each edge has five grid accumulations greater than this value and repeat four times. The data recorded are the result of the simulation for this group.

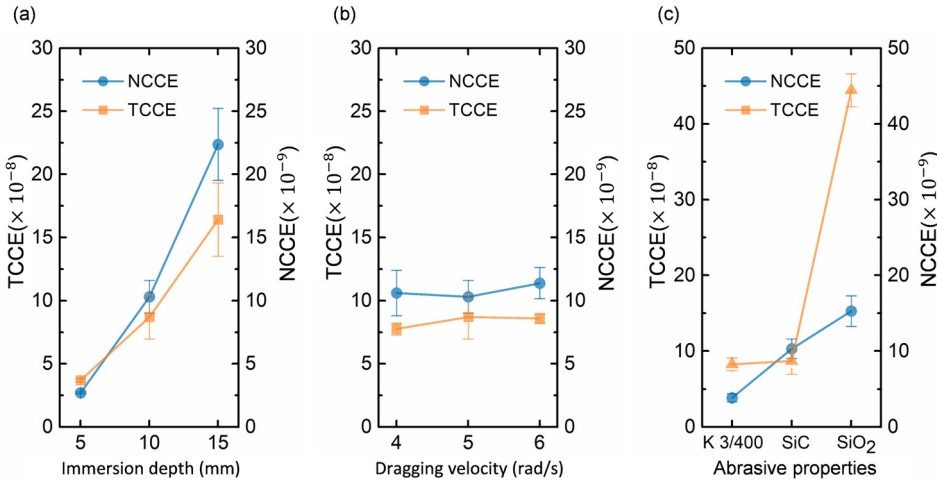

**Figure 10.** Effect of (**a**) Immersion depth, (**b**) Dragging velocity, and (**c**) Abrasive properties on cumulative energy.

Figure 10 shows that both NCCE and TCCE show significant growth with increasing depth of immersion, but NCCE shows a higher rate of growth than TCCE. Changing the abrasive grains affects both cumulative energies in a more complex manner. It is evident from Equation (6) that the density and particle size of the abrasive particles affect abrasive wear. The TCCE values for K3/400 and SiC are similar due to the larger diameter yet lower density of K3/400 particles compared to SiC particles. Additionally, $SiO_2$'s high hardness and larger particle size contribute to its significantly greater TCCE compared to NCCE. The dual effect can be explained by the two material removal formulas, Formulas (13) and (6), mentioned earlier. Furthermore, NCCE shows a uniform increase from K3/400 to $SiO_2$,

but its rate of increase is considerably lower than that of TCCE. The influence of dragging velocity on cumulative energies is insignificant.

### 4.2. Analysis of Variance

The results are analyzed using the Taguchi analysis method based on the data in Table 4, thus verifying the simulation experiment. Figure 11 depicts a comparison between the cumulative energy obtained from the simulation and the experimentally observed edge radius, while Figure 12 demonstrates the influence of different factors on the K value. Variance analysis tables for factors affecting the K value and edge radius are shown in Tables 5 and 6, respectively.

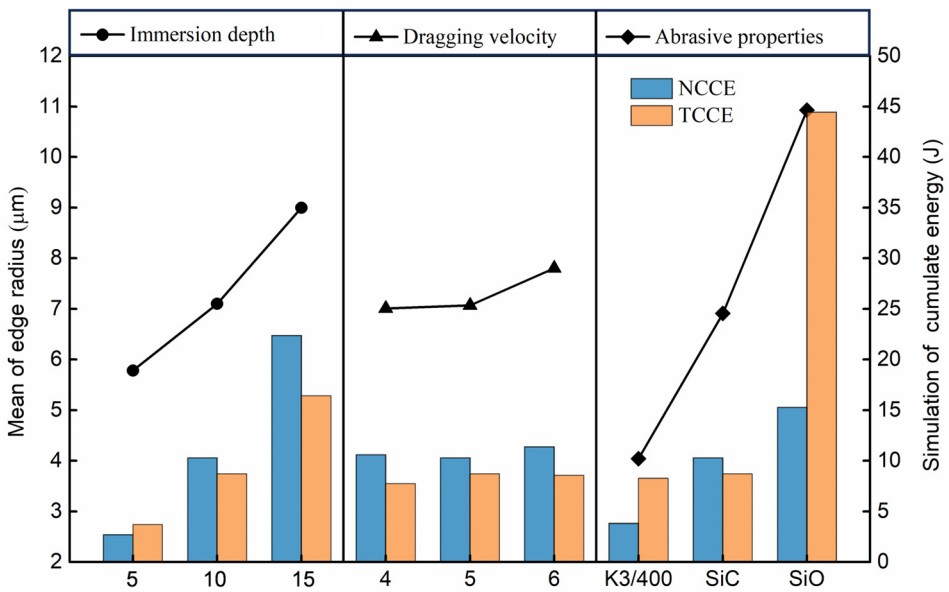

**Figure 11.** Result of mean of radius after DF in the simulation and the experiment.

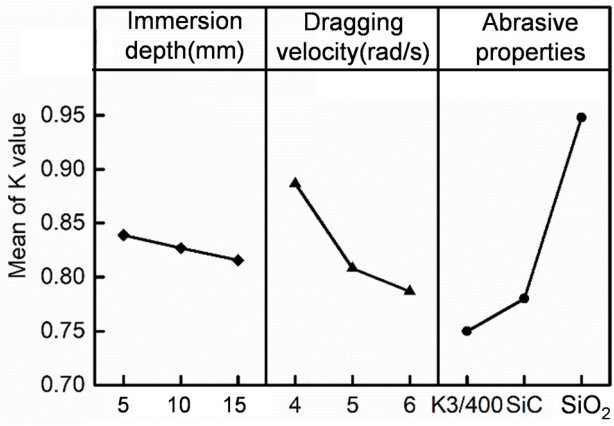

**Figure 12.** Result of mean of K value after DF in the experiment.

**Table 5.** ANOVA analysis and F-test results for the edge radius.

| Factors | Sum of Square | Degrees of Freedom | Mean of Square | F Value | *p* |
|---|---|---|---|---|---|
| Immersion depth | 15.207 | 2.000 | 7.604 | 7.031 | 0.049 * |
| Dragging velocity | 1.156 | 2.000 | 0.578 | 0.534 | 0.623 |
| Abrasive properties | 71.693 | 2.000 | 35.847 | 33.146 | 0.003 ** |
| error | 4.326 | 4.000 | 1.081 | | |

"*" means that the results are generally significant, "**" means that the results are very significant.

**Table 6.** ANOVA analysis and F-test results for the K value.

| Factors | Sum of Square | Degrees of Freedom | Mean Square | F Value | $p$ |
|---|---|---|---|---|---|
| Immersion depth | 0.0008 | 2.0000 | 0.0004 | 0.3466 | 0.726 |
| Dragging velocity | 0.0167 | 2.0000 | 0.0083 | 7.0691 | 0.048 * |
| Abrasive properties | 0.0673 | 2.0000 | 0.0336 | 28.5277 | 0.004 ** |
| Error | 0.0047 | 4.0000 | 0.0012 | | |

"*" means that the results are generally significant, "**" means that the results are very significant.

Figure 11 shows that the abrasive type has the most significant effect on the edge radius, followed closely by the depth of immersion. In contrast, the effect of drag speed is negligible. Among the various abrasive types, $SiO_2$ produces the largest edge radius, while K3/400 produces the smallest. A significant increase in edge radius is obtained by increasing the depth of immersion from low to high. Comparing the observed trends in edge radius variation from experiments with the trends in the cumulative energy variation from the simulations demonstrates the correlation between material removal in the experiments and cumulative energy in the simulations. This confirms the effectiveness of the simulation experiments in predicting material removal rates based on cumulative energy. The changes in the edge radius are interpreted in accordance with the cumulative changes in energy observed in the simulation.

Tables 5 and 6 present the variance analysis of edge radius and K value using results from the Taguchi experiment, indicating the influence of various factors on the edge radius. The individual factor's impact on the edge radius is then compared to the overall error in the orthogonal experiment to determine their significance. A factor with a $p$-value less than 0.05 is considered significant, while $p$-values below 0.01 indicate extremely high significance. Table 5 shows that altering the type of abrasive significantly affects the edge radius, while immersion depth also has a notable impact. However, finishing velocity does not show any significant influence. The selected low values for the speed variable during the orthogonal test could be the reason for this outcome. To summarize, immersion depth and abrasive properties exhibit a significant effect on material removal rate. The impact of immersion depth has typically been disregarded in prior research, with more emphasis placed on factors such as dragging velocity and abrasive types.

Analysis of the data in Table 6 shows that the dragging velocity has a significant effect on the K value of the edge, while the abrasive properties have an extremely high significance. Figure 12 demonstrates that with an increase in immersion depth, the K value experiences a consistent linear decrease; however, the magnitude of change is trivial. An increase in the dragging velocity results in a more pronounced decrease in the K value compared to the immersion depth. Furthermore, the impact of various abrasive materials on the K value of the edge is more noticeable. The application of $SiO_2$ in DF achieves the highest K value, which approaches 1, suggesting an almost symmetrical edge arc. On the other hand, DF with K3/400 results in the lowest K value of 0.74, which is a waterfall-shaped cutting edge.

The analysis shows that the drag velocity does not significantly affect the K value or the radius of the cutting edge compared to the other two parameters. Thus, it is necessary to investigate the particle velocity. The velocity directly affects the contact mechanics through the strain rate in the Archard equation that describes sliding wear. Despite the formula, both simulation and experimental results indicate that the influence of dragging speed is not significant. The simulation results for each group should be used to identify, extract, and arrange the motion velocity of particles with the highest speed. Figure 13 shows the velocity curves for the three groups. The curves indicate that only a few particles move at a speed similar to that of the tool. It can be deduced that these particles are situated nearest to the tool and are responsible for material removal. In the Archard equation, it is necessary to consider the relative speed of the collision process between the abrasive particle and

the milling tool, which is much lower than the actual dragging speed. This section of the particle is minimally affected by the tool's speed, resulting in the negligible impact of drag velocity on the above outcome.

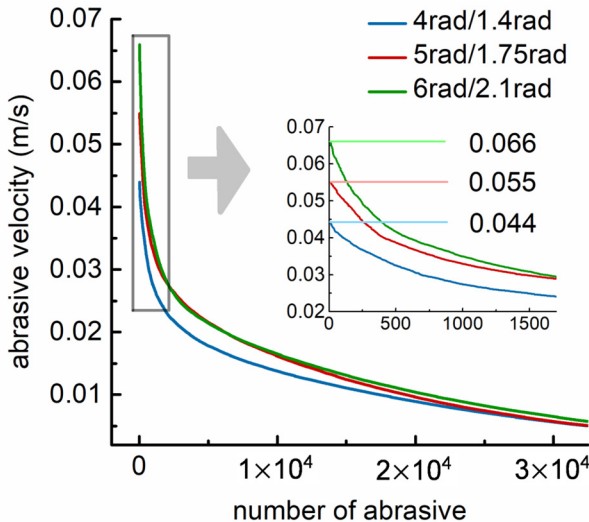

**Figure 13.** Abrasive velocity in DF.

## 5. Conclusions

This paper explores the abrasive motion during DF and the mechanism of material removal, proposing a combined form of erosive and abrasive wear. The DEM models to replicate milling tool DF under varying parameters are established. The effects of process parameters on the material removal rate caused by the two forms of wear are analyzed. The impact of abrasive properties, immersion depth, and dragging velocity on the edge radius and K value of the milling tool during DF are explored through orthogonal experiment. The following conclusions are obtained:

1.  The alteration of process parameters has resulted in a significant change not only in the cutting edge radius but also in the K value, which previous studies have ignored. Previous articles suggested that the K value is only affected by the ratio of forward and backward rotation. The K value is significantly impacted by both abrasive properties and dragging velocity. The influence of abrasive type, dragging velocity, and immersion depth on the K value is ranked from high to low. As the dragging velocity increases, the K value decreases.

2.  Based on the results of the DEM, the distribution of TCCE and NCCE differ in the position of the cutting edge and caused a different distribution of material removal. The theoretical analysis also shows the sensitivity of immersion depth, wear particle type, and wear particle velocity to the removal rate of the two wear forms. This is the direct cause of the varying K values in conclusion 1.

3.  From the range analysis of the orthogonal experiments, both the immersion depth and the drag velocity have a positive effect on the material removal rate, but the positive effect of the drag velocity is not clear in the drag finishing. By comparing the abrasive velocity and the tool velocity in the simulation results, it is found that the change in the drag velocity is weakened in the relative velocity change of the tool and the abrasive particle, causing the effect to be insignificant.

**Author Contributions:** The first author L.Z. has been responsible for writing this paper, designing the experimental process, and analyzing the experimental results. D.L. has been responsible for collecting experimental data and the examination. The corresponding author Y.W. has been responsible for determining the overall logical structure of the paper and guiding the entire experiment. All authors have read and agreed to the published version of the manuscript.

**Funding:** This research received no external funding.

**Data Availability Statement:** Data are contained within the article.

**Conflicts of Interest:** Author Dejin Lv was employed by the company Guohong Tool System (Wuxi) Co., Ltd. The remaining authors declare that the research was conducted in the absence of any commercial or financial relationships that could be construed as a potential conflict of interest.

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
