# Peer review of "Research on Cutting Edge form Factor of Milling Tool after Drag Finishing Preparation Based on Discrete Element Method"

_machines, doi:10.3390/machines12040236_

Round 1

Reviewer 1 Report

Comments and Suggestions for Authors

I think the article is well prepared. The choice of methods is adequate, and the results seem to be supported by the measurements. I have a few suggestions that would improve the article before publication in the Machines journal:

1.               How is the cutting tool selected? Explain the selection.

2.               Why is the selected tool representative of your research?

3.               Because the Taguchi method was used, the following questions arise:

-            Taguchi's experimental design does not allow for the assessment of control factor interactions. Have you neglected them, or are they demonstrably non-existent?

-            Were the regression equations determined on the basis of ANOVA? Is there any correlation between the regression equations and those determined from the theoretical analysis of your concept?

-            Was a confirmation test performed to verify the predicted values compared to the experimental values?

4.               Improve conclusions. Conclusions should be related to the results obtained from the investigation.

Minor mistakes:

Line 194 – radio?

Line 195 – The relation …

Line 240 - cutting

Reviewer 2 Report

Comments and Suggestions for Authors

comment in the attachment.

Author Response

Please refer to the attached file:

Reviewer 3 Report

Comments and Suggestions for Authors

In this paper, the effect of drag parameters on material removal of tool edge is investigated by theoretical modeling, DEM simulation and Taguchi experiments. However, the theoretical and DEM simulations are so different from the actual working conditions that the experiments cannot provide a robust validation of any model, and the proposed model presents limited novelty on its own.

1.          The authors developed an erosive wear model and an abrasive wear model under the action of single grain. However, the abrasive motion in real working conditions is so complex that it is difficult for the model to provide any quantitative guidance for the experiments. The model accuracy lacks sufficient validation. For example, are there other forms of material removal during finishing such as friction, plowing, and cutting removal? How to determine the percentage of erosive and abrasive wear?

2.          Was the removal of tool material taken into account in the DEM simulation or was the analysis based only on the collision model? If the tool material can be removed, how are the failure criteria and the intrinsic model of the material defined? Some details should be highlighted, e.g., Figure 4(a) abrasive velocity distribution should provide a localized magnified view. It should be considered how the velocity vector is related to the wear model?

3.          Note that the abrasive wear model is not subjected only to normal loads. NCCE and TCCE are not significantly associated with erosive and abrasive wear.

4.          The experimental results in Figure 9 should provide error bars. How are the error bars for the simulation results in Figure 10 obtained? In Figure 5, the experimental setup is obscured and more details should be shown.

Author Response

Please refer to the attached file:

Round 2

Reviewer 2 Report

Comments and Suggestions for Authors

The authors make corrections to the article. Thank you. I recommend the article for publication in the journal.

Reviewer 3 Report

Comments and Suggestions for Authors

The author has made revisions according to the review comments.